# WFO: Cloud-Edge Cooperative Data Offloading Strategy Akin to Water Flow

Shaonan Li [†], Yongqiang Xie *,[†], Zhongbo Li, Jin Qi, Junjie Xie and Zexin Yan

Institute of System Engineering, Academy of Military Sciences, Beijing 100010, China;
leeshaonan@outlook.com (S.L.); zbli2021@outlook.com (Z.L.)
* Correspondence: yqxie2021@outlook.com
† These authors contributed equally to this work.

**Abstract:** The exponential growth of video data in networks has led to video flow occupying a significant proportion of network traffic, causing congestion and poor service quality. To address this issue, it is crucial to quickly offload data and ensure high-quality service for users, especially in the context of cloud-edge collaboration. We propose a strategy for collaborative data offloading between cloud and edge computing, analogous to water flow (WFO). When users simultaneously access the same data from the same data source, WFO can serve more users within the limited bandwidth of the cloud while maintaining the quality of service. WFO creates a water flow-like data link between nodes to enable data offloading, using multiple nodes in collaboration to offload data for a single node. Experimental results show that compared with typical methods, such as fair-queue and first-come-first-served, WFO can significantly reduce the data offloading delay, guarantee service quality, and effectively reduce network congestion. Moreover, the number of service nodes can be as numerous as possible.

**Keywords:** cloud computing; cloud-edge cooperative; data offloading; FQ; FCFS

## 1. Introduction

Data offloading of cloud-edge collaboration has been the focus of cloud- and edge-computing in recent years. As video data on the internet increases, traditional cloud computing needs improvement to ensure user service. However, the emergence of edge computing compensates for the deficiencies of cloud computing. The data offloading strategy of cloud-edge collaboration can serve as many users as possible while ensuring service quality, especially in situations with limited network bandwidth. The cloud-edge collaboration will be the central technology in upcoming networks [1]. The data offloading strategy of cloud-edge collaboration is a vital component of this advancement.

Cloud computing has emerged as a prominent industry in information technology due to its ability to offer a diverse range of services. It is considered a vital resource pool with elastic and flexible applications, dynamic extensibility, and a high availability of performance and hosting services. Over the past decade, the academic and industrial fields have extensively researched cloud computing. According to reports by IDC and CNNIC, internet access terminals have exceeded 50 billion, with video users accounting for 94.5% [2]. As the number of network users continues to grow, more than cloud computing's data offloading capacity is required for services that require real-time performance. Furthermore, uploading and offloading video streams strain the cloud data center's network bandwidth. Edge computing was developed to address the issues of data transmission delays and bandwidth consumption in cloud computing. In cloud and IoT services, downstream data refers to cloud services, while upstream data refers to IoT services [3]. By leveraging the proximity of users to edge computing nodes, services can be obtained from the edge, leading to a significant reduction in data processing delays and bandwidth consumption associated with cloud computing. With the growing research on edge computing, it has

become apparent that cloud-edge collaboration will emerge as a new computing mode and a promising area for further investigation [1,4].

Cloud-edge collaboration [5] has widened in many directions, such as computation offloading [6,7], data offloading [8,9], data sharing [10,11], resource allocation [12,13], and privacy protection [14]. Data collaboration is a crucial research area in cloud-edge collaboration. Data offloading is indispensable for data collaboration [3]. In cloud-edge collaboration architecture, the user prioritizes service quality over the source of the service, whether it is from the cloud or edge computing. Edge computing addresses the issue of cloud computing by bringing the service closer to the user, thereby reducing the distance between users and the service. Integrating cloud and edge computing in a collaborative computing mode can alleviate the operational pressure of cloud data centers and enhance the reliability of network services. Cloud data centers must swiftly transfer data to edge nodes to disseminate video data throughout the network. Hence, exploring data offloading in cloud-edge collaboration can leverage the benefits of both cloud and edge computing to deliver superior services to users while reducing the congestion probability of the central node, increasing the data uninstallation rate, and reducing user delays [2].

According to a 2018 report on *Global Internet Phenomena*, video data makes up nearly 58% of downstream network traffic [15]. Due to its large size, video flow requires more bandwidth and transmission time than other data types, which significantly strains cloud data centers, often leading to congestion issues [16].

The conventional cloud-computing service model provides users with fair queue (FQ) [17,18] and first-come-first-served (FCFS) strategies. These strategies were initially designed for scheduling purposes but have evolved into a reliable data-offloading strategy in our current application scenario. FQ [19–24] allocates an equal bandwidth to each user based on their current bandwidth usage, while FCFS [25–27] generates a service stack based on a user's time series and follows the principle of first come, first served. These two offloading models are well-established and commonly used in modern cloud networks, forming the foundation of the Internet. While FQ can offer services to all users simultaneously, it has limitations. As the bandwidth is fixed, increasing the number of users can result in congestion, which means that not all users can request services [28,29]. Additionally, FCFS has its limitations, such as a small number of users and long response times for users [30].

In this paper, we propose a data offloading strategy that can be implemented in both traditional cloud computing architecture and cloud-edge collaboration architecture. This strategy involves grouping users who have not requested services with nodes that have requested services, resulting in multiple link groups for data offloading. This approach allows for increased user requests that can be handled simultaneously. Once the service request of a preceding node is completed, it collaborates with subsequent nodes to offload the data, thereby reducing delays.

WFO enables multiple users to access the same data in a data center simultaneously, with three main features. Firstly, edge nodes are queued based on their time series and allocated bandwidth according to their actual requirements until no more bandwidth is available for the cloud. Nodes that come after are then mounted onto the pre-existing nodes within the user's tolerance delay to create multiple packet links. Second, the priority of the edge nodes is calculated within the group, and data offloading is sorted accordingly. Once the priority level is determined, data offloading and segmentation are based on this arrangement. The data then flows through the arranged nodes to the next node. When the first node stops receiving data, the cloud offloads the data to subsequent nodes. Third, data offloading and segmentation are performed as per the arrangement. The data flows through the arranged nodes to the next node. Once the first node stops receiving data, the cloud offloads the data to the subsequent nodes.

We conducted this study due to the need for data offloading in cloud-edge coordination networks and customer requirements for high-quality services. **The motivation can be summarized as follows:**

- Use the limited bandwidth resources of the cloud and edge to provide services for more users when the number of users increases significantly. When offloading time-sensitive data, ensure data validity and reduce offloading delay. Reduce the usage of cloud and edge infrastructure resources using a data offloading strategy for cloud and edge collaboration.
- Based on historical experience, when many users suddenly access a data center, the bandwidth faces tremendous pressure. Users often have to wait for services. For example, in live broadcasts of large-scale sports events, the user terminal often appears to stall and pause. This is because physical resources limit the number of services available to the users. This can be improved by adding physical infrastructure. However, there are significant costs involved. We hope to present innovations for algorithms and system architectures to maximize system performance under a limited bandwidth.
- With an increase in video traffic in the network, the timeliness of data must be considered. The timeliness of data is related to the user's service experience. Therefore, delay in data offloading was a crucial factor in this study.
- Under the development trend of cloud-side collaboration, the traditional data offloading method cannot provide its advantages in such an architecture. Data offloading is the foundation of cloud-edge collaboration. However, there is still little research on data offloading, in which data flows are sent from the data center to the edge nodes. A strategy must be designed and developed to leverage cloud- and edge-resources to improve network performance.
- Currently, cloud edge collaboration technology is an important component of next generation networks. Since the application of edge computing is not yet fully developed, the research of using cloud-edge collaboration to accomplish data offloading is not yet deep. Most of the current studies consider data offloading on the cloud-side or end-side.

This paper focuses on data offloading as a critical technology for cloud-edge collaboration. In order to ensure a user's QoS in the network architecture of cloud-edge collaboration, a performant cloud and edge computing are necessary. However, the existing data offloading model needs to be revised to meet these requirements, and a new data offloading strategy is needed to adapt to cloud-edge collaboration. Due to the large amount of video data, network users tend to utilize the data center's bandwidth for extended periods, which can result in some users not receiving data and questioning the reliability of the cloud. Therefore, minimizing the delay gap between users is crucial to our work.

**In summary, we make the following contributions:**

- A cloud-edge collaborative data offloading strategy is proposed. WFO can serve more users with the limited outgoing bandwidth of the source node and ensure the delay requirements of users.
- We analyze the limitations of the traditional data offloading model in the cloud-edge collaboration architecture. The application scenario of multiple users accessing the same data in the same node is modeled.
- We evaluate WFO's sensitivity to cloud-edge collaboration data offloading challenges such as the number of user services, the average delay of data, and the discrete distribution of delay.

This paper is organized as follows. Section 3 introduces the system model and the problem, and composes the framework of the paper. Section 2 introduces the related works of cloud-edge collaborative data offloading strategies and two typical data offloading models, FQ and FCFS, followed by some discussions. Section 4 introduces the data offloading strategy WFO which can greatly increase the number of users while ensuring the quality of service. Experimental results are presented in Section 5 and concluding remarks in Section 6.

## 2. Related Work

*Latency*. Zhao et al. [31] considered a data transmission network architecture based on the Manhattan mobility model and proposed a temporal convolutional network (TCN) model to predict the allocation of the weight of delay. Finally, they solve the optimal routing problem using a genetic algorithm based on a reinforcement learning mechanism (RLGA) to pre-allocate resources for offloading requests. O.B. Yetim et al. [32] proposed four schemes to dynamically and adaptively reduce the delay in user services. These schemes (Adaptive, Decision Tree-Based, Hybrid, and Lazy) have low-overhead and are effective. Ndikumana et al. [33] proposed a new communication planning approach that enables the vehicle to optimally preselect the available RATs such as Wi-Fi, LTE, or 5G to offload tasks to clouds when its local resources are insufficient.

*Engergy*. Zhou et al. [8] proposed a hierarchical multi-agent deep reinforcement learning (H-MADRL) framework to maximize the overall energy efficiency by jointly optimizing the beamforming of the access point (AP) and the users' offloading decisions.

*Sum of successfully offloaded tasks*. He et al. [34] proposed a task flow graph (TFG) to depict the collaborative data offloading procedure in earth observation satellite networks. Based on the proposed TFG, authors formulate the studied problem as integer linear programming (ILP) to maximize the weighted sum of successfully offloaded tasks.

*QoS*. Majumder et al. [35] present a novel data offloading scheme built upon the exponential learning-based minority game (MG) theory which leverages the unlicensed Wi-Fi spectrum for the cellular users while at the same time, maintains the quality of service.

*Throughput*. Jia et al. [36] proposed a collaborative data offloading scheme to maximize the overall system throughput. The data amount of each offloading task is represented as the number of equal packets. As such, they introduced decision variables to indicate whether a larger number of packets are scheduled instead of which one task is offloaded, thereby significantly enlarging the solution space and further increasing the computation complexity. B.H. Jung et al. [37] considered a user-centric Wi-Fi data offloading model in heterogeneous networks and optimized the user throughput. The heterogeneous network collects network information, such as the number of users in Wi-Fi networks and their traffic load. Then, the network decides the specific portion of traffic to be transmitted via Wi-Fi networks. Spangelo et al. [38] devised an iterative algorithm by progressively tightening the realistic constraints of data offloading to maximize the total amount of earth observation data.

*Data loss*. Parmeet [39] proposed a technique for the fault tolerant offloading of data by IoT devices such that the data collected by them is transferred to the CC with minimal loss. The proposed technique employs opportunistic contacts between IoT and mobile fog nodes to provide a fault tolerant enhancement to the IoT architecture.

We summarize the existing works on computing offloading in Table 1.

**Table 1.** Comparison of existing data offloading algorithms.

| Optimization Objective | Reference | Strategy | Processing Node | | | Offloading Type |
| --- | --- | --- | --- | --- | --- | --- |
| | | | End | Edge | Cloud | |
| | O.B. Yetim et al. [32] | Adaptive Decision Tree-Based Hybrid and Lazy | √ | | | Delay tolerance |
| Latency | Zhao et al. [31] | Heuristic & RL | √ | | √ | Allocate resources |
| | Ndikumana et al. [33] | RL | √ | | √ | RATs preselect |
| Engergy | Zhou et al. [8] | DRL | √ | √ | | AP decision |
| Success rate | Spangelo et al. [38] | Heuristic | | | √ | Path decision |

**Table 1.** *Cont.*

| Optimization Objective | Reference | Strategy | Processing Node | | | Offloading Type |
| | | | End | Edge | Cloud | |
|---|---|---|---|---|---|---|
| Throughput | Jia et al. [36] | Iterative optimization | √ | √ | | Joint scheduling |
| | Jung et al. [37] | trust-region-dogleg search algorithm | √ | | | WiFi offloading |
| | Spangelo et al. [38] | Iterative optimization | √ | | √ | Communication scheduling |
| Data loss | Parmeet [39] | Heuristic | √ | | √ | Fault tolerant |
| QoS | Majumder et al. [35] | Game theory | √ | | | QoS |
| WFO | Our proposed | Heuristic | | √ | √ | Content delivery |

## 3. System Model and Problem Statement

As illustrated in Figure 1, we consider static system integrated edge computing and cloud computing nodes. Each of the edge computing nodes is equipped with a data server to provide data services to a community. Take a large video conference as an example. At one point, an online video conference was organized on the cloud node, and participants accessed the conference through their own edge computing nodes. The minutes of the video conference and the dynamics of all participants during the conference are stored on the cloud in the form of video files. A large amount of video data will be generated. After the conference, the conference video needs to be copied to edge computing nodes for backup, in order to facilitate review and let non-participants watch. **Therefore, the problem of this paper is how to quickly realize the backup of cloud data on many edge nodes through cloud-edge collaboration.** Then, we elaborate the related parameters, delay model, and overview of the framework.

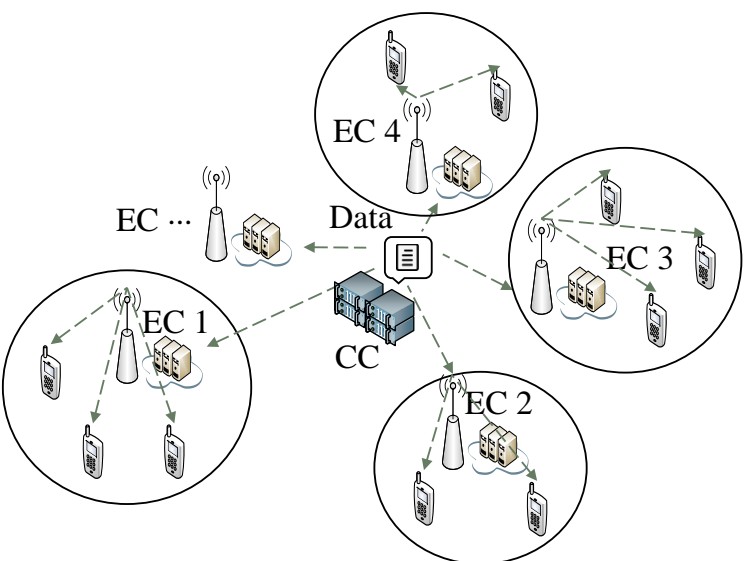

**Figure 1.** An illustration of the data offloading scenario integrated edge computing and cloud computing nodes.

### 3.1. Related Parameters

The parameters used to represent the model shown in Figure 1 include data parameters, transmission parameters, and bandwidth parameters. Data parameters include data size $S$ and data tolerance delay $\tau'$ measured by MB and ms, respectively. Transmission parameters, including transmission delay $\tilde{\tau}_{i,j}$ (in ms) and forwarding delay $\hat{\tau}_i$ (in ms), indicate the delay of the data stream from one node to another and the delay of data stream forwarding on the node. The bandwidth parameters include the bandwidth of the cloud $\mathcal{B}$ (in MB/s), bandwidth between the edge node and the cloud $B_{i,C}$ (in MB/s), and bandwidth between

edge nodes $B_{i,j}$ (in MB/s). These important notations used throughout this article are listed in Table 2.

**Table 2.** Summary of Notations and Definitions.

| Notations | Definition |
|---|---|
| **System Model** | |
| $C$ | $\triangleq (S, \mathcal{B})$, a cloud |
| $\gamma$ | The number of ECs that obtain data directly from the cloud. |
| $\mathcal{B}$ | The bandwidth of the cloud. |
| $\mathcal{L}_k$ | The $k$th data forwarding link. |
| $\mathcal{N}$ | The number of ECs. |
| $\mathcal{L}'_k$ | The $k$th data forwarding link that has been prioritized. |
| $S$ | The amount of data needed to be offloaded in the cloud. |
| $n_k$ | The number of nodes in $\mathcal{L}_k$. |
| $i, j$ | $\in [0, n]$, the $i$th and $j$th ECs. |
| $\alpha_{i,j}$ | $\in \{0, 1\}$ represents the priority between nodes $i$ and $j$. If the priority of $i$ is greater than that of $j$, $\alpha = 1$; otherwise, $\alpha = 0$. |
| $\widetilde{\tau}_{i,j}$ | The transmission delay from the $i$th node to the $j$th node. |
| $P_i$ | $\in [0, \infty)$, the priority of node $i$ in $\mathcal{L}_k$. |
| $\widehat{\tau}_i$ | The forwarding delay of the $i$th node. |
| $T_{\mathcal{L}_k}$ | The maximum data delay of $\mathcal{L}_k$. |
| $\tau'$ | The tolerate delay. |
| $\zeta$ | The $\zeta$th node in $\mathcal{L}'_k$. |
| $B_{i,C}$ | The bandwidth from node $i$ to $C$. |
| $t_\zeta$ | The data delay of the $\zeta$th node receiving all data. |
| $B_{i,j}$ | The bandwidth from node $i$ to $j$. |
| $t'_\zeta$ | The duration for the high-priority node to forward data to the $\zeta$th node. |
| $t''_\zeta$ | The duration for the $\zeta$th node to obtain data through cooperative forwarding. |
| $V_c^\zeta$ | The amount of data forwarded to the $\zeta$th node through cooperative forwarding. |
| $V_r^\zeta$ | The amount of data forwarded to the $\zeta$th node through the high-priority node. |
| $\theta, \lambda, \vartheta, \epsilon$ | The auxiliary parameters used in the proof. |
| $V'_{\rho,\zeta}$ | The amount of data that node $\rho$ forwards to node $\zeta$. |

### 3.2. Delay Model

We assume that the network state is relatively static during data forwarding. The bandwidth, transmission delay, and forwarding delay do not change dramatically.

**Data delay**: The delay for a node to receive all data. Therefore, for an edge node, the delay to receive all the data from cloud can be expressed as:

$$T_i = \frac{S}{B_{i,C}} \tag{1}$$

If part of the data of node $i$ comes from the cloud and the other part comes from node $j$, then the delay to receive all the data can be expressed as:

$$T_i = \frac{S}{B_{i,C} + B_{i,j}} \tag{2}$$

If all the data received by node $i$ is forwarded by node $j$, then the data delay is calculated as:

$$T_i = \frac{S}{min(B_{j,C}, B_{i,j})} \tag{3}$$

**Link delay**: The delay of data stream forwarding from the cloud to an edge node through other nodes. Thereby, the link delay can thus be calculated as:

$$\tau = \sum_{i,j \in C \cup \mathcal{L}} \widetilde{\tau}_{i,j} + \sum_{i \in \mathcal{L}_k} \widehat{\tau}_i \tag{4}$$

where $\mathcal{L}$ represents a data forwarding link.

### 3.3. Overview of Framework

WFO is designed to effectively use cloud and edge nodes to realize fast same data forwarding under the service architecture of cloud-edge collaboration. Specifically, the approach is meant for live streaming. This is a shift from a traditional one-to-one transmission to a one-to-many transmission. The design process of WFO consists of three steps. *First, grouping nodes.* Edge nodes are queued and numbered in chronological order of request. The bandwidth of the cloud is allocated based on the node ID and the actual bandwidth requirements. Until the cloud has no bandwidth left, the subsequent nodes are mounted on the earlier nodes within the user's tolerance delay. Multiple data forwarding links are formed. Data forwarding between groups does not affect each other. *Second, calculating priorities.* When grouping is performed, the flow of data between nodes is not determined. We must calculate the priority of nodes in a group and sort the data routing based on priority. *Finally, data offloading.* Data are transmitted in the permutation generated in the second step, passing through permutation nodes, akin to water flow, to the next node. When the first node receives the data, the first node and cloud offload the data to subsequent nodes until the last node receives the data. The overall framework is shown in Figure 2.

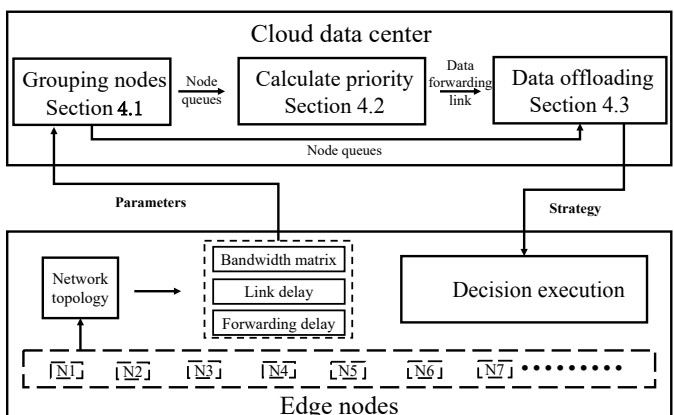

**Figure 2.** WFO sends the parameters generated by the edge nodes to the cloud data center. The strategy will be sent to VMS for execution through a three-step generation strategy: grouping nodes, calculating priority, and data offloading.

## 4. WFO Design

### 4.1. Grouping Nodes

The primary function of grouping nodes is to increase the number of nodes so that more users can receive data simultaneously. The execution process is as follows. Cloud sorts and numbers the edge nodes in chronological order. Cloud allocates the outgoing bandwidth to the edge nodes. Edge nodes that do not request the bandwidth are mounted on the preceding nodes to form multiple data links. The number of nodes on each link is determined by the tolerance time. In other words, the delay of the data arriving at the last node on each link cannot exceed the maximum tolerable delay. For delay-sensitive data, the larger the delay, the smaller the value of the information [40]. In order to avoid data invalidation, the tolerance delay is set. The setting of the delay tolerance depends on the characteristics of the application and data.

We assume that there is a group of edge nodes $\mathcal{N}$ that can be served by a cloud $C \triangleq (S, \mathcal{B})$. $S$ and $\mathcal{B}$ represent the amount of data needed to be offloaded in the cloud and the bandwidth of the cloud, respectively. All edge nodes access the same data on the cloud. We use $N_i(i \in [1, |\mathcal{N}|])$, and $B_{i,j}(i, j \in [1, |\mathcal{N}|])$ represents the $i$th edge node and the bandwidth between the $i$th edge node and $j$th edge node, respectively. Figure 3 illustrates the execution process of grouping nodes. Assume that the cloud outgoing bandwidth is 80 $MB/s$. The bandwidth between each node and the cloud is assumed to be 20 $MB/s$. If there are 50 edge nodes requesting data, cloud will number the edge nodes. The cloud serves edge nodes according to the numbering order. According to this hypothesis, only nodes $N_1$, $N_2$, $N_3$, and $N_4$ can request data directly from the cloud. The other nodes have to wait. To improve efficiency, we connect $N_5$ to $N_1$, and $N_6$ to $N5$. This results in a data forwarding link include edge nodes $N_1$, $N_5$, and $N_6$, as shown in Figure 3. We use $\mathcal{L}_k$ ($\mathcal{L}_1 \triangleq N_1, N_5, N_6$), which represents the $k$th data forwarding link. Therefore, $N5$ and the subsequent nodes can obtain data through the first four nodes. The number of nodes in each link depends on the delay in data arriving at $N_6$, which means that the delay in data arriving at $N_6$ cannot exceed the tolerable delay. Therefore, the data forwarding link must meet the following conditions:

$$\sum_{i,j \in C \cup \mathcal{L}_k} \widetilde{\tau}_{i,j} + \sum_{i \in \mathcal{L}_k} \widehat{\tau}_i \leq \tau' \tag{5}$$

where $\widetilde{\tau}_{i,j}$ is the transmission delay from the $i$th edge node to the $j$th edge node. $\widehat{\tau}_i$ is the forwarding delay of the $i$th edge node. $\tau'$ is the tolerate delay. The number of nodes in each data link can be determined by Formula (5).

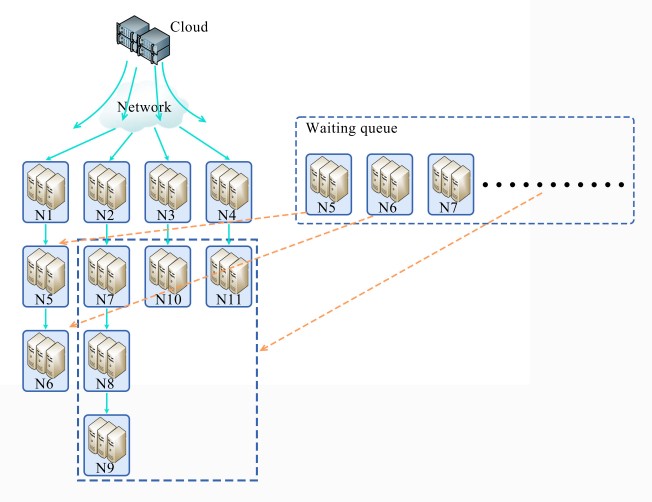

**Figure 3.** Edge nodes request data according to FCFS policies. The waiting node is mounted on the preceding node. Finally, nodes are grouped into multiple data forwarding groups.

We designed Algorithm 1 ***Grouping nodes*** to generate data forwarding links. The algorithm takes as input of the bandwidth between the $i$th and $j$th nodes $B_{i,j}$, the bandwidth between the $i$th node and cloud $B_{i,C}$, the bandwidth of the cloud $\mathcal{B}$, the forwarding delay $\widehat{\tau}_i$, the link delay $\widetilde{\tau}_{i,j}$, and the tolerance delay $\tau'$. The output shows multiple data forwarding links $\mathcal{L}_k$. The algorithm arranges the nodes by time, similar to the FCFS strategy, and generates an initial queue. We set an intermediate variable $\gamma$, which represents the number of nodes that get data directly from the cloud. Lines 4–11 are used to calculate the delay that data arrives at the $j$th node. If the delay is smaller than the tolerance delay, the $j$th node is added to $\mathcal{L}_i$ and continues to cycle. If the delay is greater than the tolerance delay, the $j$th node is added to $\mathcal{L}_i$, then breaks.

---

**Algorithm 1:** Grouping Nodes

---

**Input:**
$B_{i,j}$, $B_{i,C}$, $\mathcal{B}$, $\widehat{\tau}_i$, $\widetilde{\tau}_{i,j}$, and $\tau'$
**Output:**
data forwarding links $\mathcal{L}_k$

1   Initialize edge nodes queue and $\gamma$
2   Calculate $\gamma$ according to $B_{i,C}$, $\mathcal{B}$
3   **for** $1 \leq i \leq \gamma$ **do**
4     $y \leftarrow \widetilde{\tau}_{i,C} + \widehat{\tau}_i$
5     **for** $\gamma + 1 \leq j \leq \mathcal{N}$ **do**
6       $z \leftarrow y + (\widetilde{\tau}_{i,j} + \widehat{\tau}_j + \widetilde{\tau}_{j,j+1})$
7       **if** $(z \geq \tau')$ **then**
8         Put $N_j$ into $\mathcal{L}_i$
9         Break
10       **else**
11         Put $N_j$ into $\mathcal{L}_i$
12         Continue
13       **endif**
14   **return** $\mathcal{L}_i$

---

### 4.2. Calculate Priority

The priority calculation occurs after the grouping nodes. The purpose of computing the priority is to further reduce the data delay. In this section, we use $T_{\mathcal{L}_k}$, $a_{i,j}$, $x_i$ to represent the maximum data delay, the priority between nodes $i$ and $j$, the priority of the $i$th node in the data forwarding link $\mathcal{L}_k$, respectively. Figure 4 shows the results of the calculated priority.

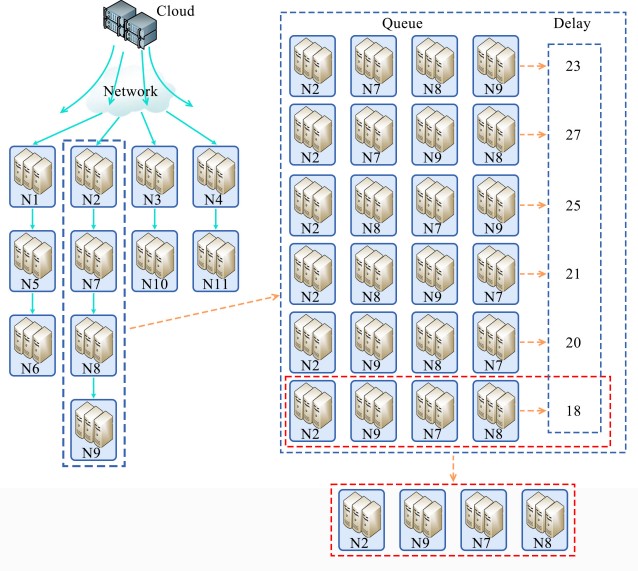

**Figure 4.** Data forwarding link priority calculation. Different data forwarding links lead to different delays of the last node. This may cause a considerable impact on data offloading performance. Priority is used to calculate the link with the shortest delay.

Computing priorities are time-consuming processes that consume considerable computing and storage resources. Despite the FCFS principles, we can still serve more users without counting their priorities. However, we still choose to do this because it dramat-

ically reduces the data delay and dispersion of data delay between nodes. The different forwarding order has a different data delay because of the difference in transmission rates between nodes. For example, there are three nodes, A, B, and C, with a bandwidth of 5 MB/s for AB, 10 MB/s for BC, and 2 MB/s for AC. For data sent by node A for 50 MB data, in the order of A to B to C, the time delay for B and C to receive all data is 5 s and 25 s, respectively. There is an average time delay of 15 s. For the order of A to C to B, the time delay for B and C is 25 s and 25 s, respectively. There is an average time delay of 15 s. For node B, the first path is preferable.

Consider a data forwarding link $\mathcal{L}_2 \triangleq \{N_2, N_7, N_8, N_9\}$ in Figure 4 as an example. The link status and data delay of the last node are shown on the right-hand side. The data delay consists of a transmission delay and a forward delay. Different forwarding sequences have a different data delay, $T_{\mathcal{L}_2}$. The delay calculated by the original queue with no priority is 23 ms, which is clearly suboptimal. Link $\{N_2, N_9, N_7, N_8\}$ has a delay of 18 ms, which is the best forwarding link. The data are then forwarded in this path. The priorities of the nodes were $\{P_2 > P_9 > P_7 > P_8\}$. The relationship between data delay and priority can be formulated as:

$$T_{\mathcal{L}_k}(\alpha_{i,j}) = \frac{S}{B_{k,C} + \sum\limits_{i \in \mathcal{L}_k} \sum\limits_{j \in \mathcal{L}_k} \alpha_{i,j} B_{i,j}} \tag{6}$$

where $\alpha_{i,j}$ is the priority parameters between the $i$th node and $j$th node. Variable $\alpha_{i,j}$ only contributes to Equation (6) when it is equal to 1, which means that the priority of $i$ is greater than that of $j$. The priority parameters $\alpha_{i,j}$ can be represented as in Table 3. The horizontal and vertical axes in the table represent the same nodes in data forwarding link $\mathcal{L}_2$. The nodes in the vertical axis represent $i$ in $\alpha_{i,j}$. The nodes in the horizontal axis represent $j$ in $\alpha_{i,j}$. According to Equation (6), if the priority of $i$ is greater than that of $j$, then $\alpha_{i,j} = 1$; otherwise, $\alpha_{i,j} = 0$. For example, Node 2 has a higher priority than Node 7. The intersection of row 2 and column 3 in the table is 1, which means $\alpha_{2,7} = 1$. Therefore, there exists a set of $\alpha_{i,j}$ that minimizes $T_{\mathcal{L}_k}$.

**Table 3.** Calculation results of $\alpha_{i,j}$.

| Node | $N_2$ | $N_7$ | $N_8$ | $N_9$ | $N_{10}$ | $N_{11}$ |
|---|---|---|---|---|---|---|
| $N_2$ | 0 | 1 | 1 | 1 | 1 | 1 |
| $N_7$ | 0 | 0 | 0 | 0 | 1 | 1 |
| $N_8$ | 0 | 1 | 0 | 0 | 1 | 1 |
| $N_9$ | 0 | 1 | 1 | 0 | 1 | 1 |
| $N_{10}$ | 0 | 0 | 0 | 0 | 0 | 0 |
| $N_{11}$ | 0 | 0 | 0 | 0 | 1 | 0 |

The priority calculation problem is then stated as follows: *Given a data forwarding link* $\mathcal{L}_k$, *bandwidth* $B_{i,j}$, *and data volume* $S$, *we determine the priority of the nodes so that the maximum data delay* $T_{\mathcal{L}_k}$ *is minimized*:

$$\alpha_{i,j}^{opt} = \arg\min_{\alpha_{i,j}} \left( \sum_{k=1}^{\gamma} T_{\mathcal{L}_k}(\alpha_{i,j}) \right) \tag{7}$$

$\alpha_{i,j}$ can only reflect the priority between two nodes. However, this does not represent the priority of a node within $T_{\mathcal{L}_k}$. The priority of the nodes in $T_{\mathcal{L}_k}$ can be calculated using Equation (8).

$$P_i = 1 + \sum_{j=1}^{n} \alpha_{i,j} \tag{8}$$

where $P_i$ is the priority of node $i$ in $\mathcal{L}_k$. $n$ denotes the number of edge nodes. The constant 1 is set for all nodes with a priority greater than 0. Through calculation, we can determine the size of each node's priority and generate the final data forwarding link $\mathcal{L}'_k$.

Because of the high complexity of priority calculation, we propose a heuristic priority calculation algorithm (HPC), Algorithm 2. The priority calculation time is affected by the number of nodes in the queue. The time increases exponentially with the increase of the number of nodes. The priority calculation runs depending on the number of nodes taken from several hours to a few days to produce the optimal solutions. In the experiment, when the number of nodes in a queue is 14, the solution time of the optimal approach took 1 h and 28 min. The algorithm determines the final path by iteratively finding the node with the shortest delay as the next hop. For a sequence of ECs $\mathcal{L}_k$, when $i = 1$, lines 3–7 are executed $n$ times. When $i = 1$, lines 3–7 are executed $n - 1$ times. Therefore, if there are $n$ ECs, the total number of cycles of HPC is $(n + (n - 1) + \cdots + 1) = (n^2/2 + n/2))$ time. If the algorithm is computed on the CC, the time complexity of HPC is $O(n^2)$, where $n$ is the number of ECs. Actually, line 3–7 can be evaluated by the ECs.

---

**Algorithm 2:** Heuristic priority calculation

**Input:** $\mathcal{L}_k$
**Output:** $\mathcal{L}'_k$

1    $n \leftarrow$ The number of nodes in $\mathcal{L}_k$
2    **for** $1 \leq i \leq n$ **do**
3       **for** $j \in \mathcal{L}_k$ **do**
4          **if** $j \in \mathcal{L}'_k$ **then**
5             Delete node $j$ from $\mathcal{L}_k$
6          **else**
7             Calculate the delay of node $i$ $(\tilde{\tau}_{i,j} + \hat{\tau}_i)$
8       $\mathcal{L}'_k \leftarrow$ The node with the minimum delay
9    **return** $\mathcal{L}'_k$

---

### 4.3. Data Offloading

This section introduces the data offloading process shown in Figure 5. Data offloading involves two processes. First, the data is forwarded in a set order $\mathcal{L}'_k$ to the last node. The second is multi-node coordination, which requires all nodes to accept the control and scheduling of the data source.

Assume that the grouping contains three nodes: $\{N_1, N_2, N_3\}$. The priority $N_1$ is greater than $N_2$. $N_2$ is greater than $N_3$. At the beginning of the data offloading, the data source offloads the data to $N_1$, which immediately forwards the received data to $N_2$. See Figure 5 on the left-hand side. If the data offloading task from the cloud to $N_1$ is complete, the cloud releases the bandwidth of $N_1$ for data transmission to $N_2$. Both the cloud and $N_1$ send data to $N_2$. As shown in the middle of Figure 5, same as $N_2$, $N_3$ has more nodes that work together to offload the data. See Figure 5 on the right-hand side. As node $N_1$ receives all the data, a new problem arises, which is how to distribute the amount of data sent by higher-priority nodes. For the convenience of calculation, we reorder the nodes. We use $\zeta \in [1, n_k]$, which represents the $\zeta$th node in $\mathcal{L}'_k$. Therefore, the data delay of the first node in $\mathcal{L}'_k$ can be formulated as:

$$t_{\zeta=1} = \frac{S}{B_{\zeta=1,C}} \tag{9}$$

where $\zeta = 1$ denotes the first node of the link $\mathcal{L}'_k$.

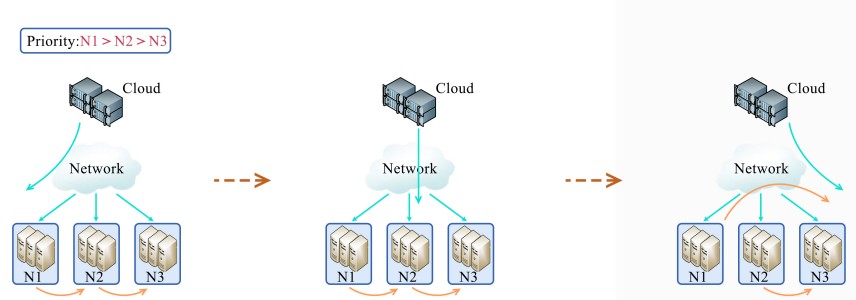

**Figure 5.** At the beginning of data transmission, data, akin to water flow, is forwarded according to link priority to ensure that more nodes can receive service responses in a short time (**Left**). When a node has received all data, the datacenters release bandwidth to unload data for other nodes (**Middle**), using multiple nodes to offload data simultaneously for lower-priority users (**Right**).

The data received by the second node consists of two parts. The first part originates from the forwarding of the first node. The transmission delay and data volume are:

$$t'_{\zeta=2} = t_{\zeta=1} \tag{10}$$

$$V_r^{\zeta=2} = t_{\zeta=1} \times (min\{B_{\zeta=1,C}, B_{\zeta=2,\zeta=1}\}) \tag{11}$$

where $t'_{\zeta=2}$ represents the duration for the first node to forward data to the second node. $V_r^{\zeta=2}$ represents the amount of data obtained by the second node in time $t'_{\zeta=2}$.

In the second part, unaccepted data is forwarded in a collaborative manner. The amount of data not received is:

$$V_c^{\zeta=2} = S - V_r^{\zeta=2} \tag{12}$$

Here, we want the same transmission delay for data not received. The data transmission delay is:

$$t''_{\zeta=2} = \frac{V_c^{\zeta=2}}{B_{\zeta=1,\zeta=2} + B_{\zeta=2,C}} \tag{13}$$

where $t''_{\zeta}$ represents the duration for the $\zeta$th node to obtain data through cooperative forwarding.

The delay of all the data received by second node is:

$$t_{\zeta=2} = t'_{\zeta=2} + t''_{\zeta=2} = t_{\zeta=1} + \frac{V_c^{\zeta=2}}{B_{\zeta=1,\zeta=2} + B_{\zeta=2,C}} \tag{14}$$

The amount of data forwarded by the first node to the second node is:

$$V'_{\zeta=1,\zeta=2} = t''_{\zeta=2} \times B_{\zeta=1,\zeta=2} \tag{15}$$

The amount of data forwarded by Node $C$ to Node $\zeta = 2$ is:

$$V'_{C,\zeta=2} = t''_{\zeta=2} \times B_{C,\zeta=2} \tag{16}$$

Recursively, the transmission delay for Node $\zeta$ in $\mathcal{L}'_2$ to obtain all the data is:

$$t_\zeta = t_{\zeta-1} + \frac{V_c^\zeta}{\sum_{\rho=2}^{\zeta-1} B_{\rho,\zeta} + B_{\zeta,C}} \tag{17}$$

where $\rho$ indicates a node with a higher priority than $\zeta$.

Because $t_\zeta$ is related to $t_{\zeta-1}$, Equation (17) can be derived as follows. See Appendix A for proof.

$$
\begin{aligned}
t_\zeta = \Big( \frac{S}{B_{\zeta=1,C}} &+ \frac{S}{(\sum_{\rho=2}^{\zeta-1} B_{\rho,\zeta} + B_{\zeta,C} - B_k')} \Big) \\
&\times \Big( 1 - \frac{B_k'}{\sum_{\rho=2}^{\zeta-1} B_{\rho,\zeta} + B_{\zeta,C}} \Big)^{\zeta-1} \\
&- \frac{S}{(\sum_{\rho=2}^{\zeta-1} B_{\rho,\zeta} + B_{\zeta,C} - B_k')}
\end{aligned}
\tag{18}
$$

where $B_k'$ is the lowest bandwidth in link $\mathcal{L}_k'$.

The amount of forwarding data undertaken by each node in collaborative forwarding is:

$$
V_{\rho,\zeta}' = t_\zeta'' \times B_{\rho,\zeta} \tag{19}
$$

where,

$$
t_\zeta'' = \frac{V_c^\zeta}{\sum_{\rho=2}^{\zeta-1} B_{\rho,\zeta} + B_{\zeta,C}} \tag{20}
$$

## 5. Experiment and Analysis

### 5.1. Experimental Setting

The results for WFO were obtained by solving the mathematical model defined in Section 4 using Python on a Linux-based 64-bit multiple-core workstation. The workstation we used had 16 cores, clocked at 2.8 GHz and 16 GB of RAM.

We considered topologies with one CC and |N| = 30, 40, 50, 60, 70, ..., 130, and 140 ECs deployed over the Internet. This topology determines two important parameters: the bandwidth matrix of the nodes to CC and the bandwidth matrix of the node to another node. The URL of the GitHub repository is https://github.com/shaonan-li/WFO (accessed on 2 May 2023).

### 5.2. Impact of the Number of Users

The experiment in this section mainly verifies the changes in WFO with respect to the number of user services. The primary consideration of the experiment was 100 ECs. The data volume is set at 300 MB. The bandwidth between the CC and ECs is generated according to the topology of the network, as recorded in the bandwidth matrix. The bandwidth was set from 3 MB/s to 10 MB/s. The link delay between nodes is set at 1–3 ms. The forwarding delay of the node is set at 1–5 ms. We assume that the data from the CC are not capped.

Figure 6 shows the number of users of the WFO strategy with the delay tolerance. The tolerance delay varied from 0–30 ms. The tolerance time was set within 30 ms because some video data are worthless when the delay exceeds 30 ms, such as live streaming. The ordinate is the number of users, which indicates the service capacity of WFO at a certain tolerance time. We considered the serviceability of WFO in five different bandwidth situations, namely 100 MB/s, 150 MB/s, 200 MB/s, 300 MB/s, 400 MB/s, and 500 MB/s. As the delay tolerance increased, the number of WFO service users increased rapidly until the maximum number of users was reached. To further illustrate this, if the tolerance time of the node is set to 3 ms, the number of users that WFO can serve under 100 MB/s is 22. This represents the number of nodes that FCFS can serve. This is the same for the other conditions. The number of users under the tolerance delay is linear until the node meets all users. Thus, we can serve more user services with minimal delay, and it has almost no impact on the user experience.

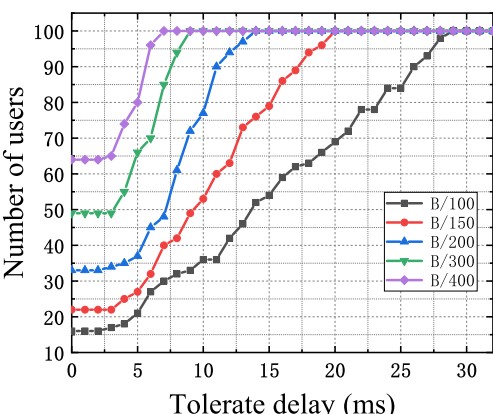

**Figure 6.** Number of users that the WFO strategy served in different tolerance times. When the tolerate delay is less than 3 ms, the number of nodes means offloading the data directly from the CC.

### *5.3. The Performance Comparison of WFO, FQ, and FCFS*

We compared the performance of WFO and the other two strategies under the conditions of varying data size, CC bandwidth, and node number concerning the average delay of offloading all data. We first show the results for *the amount of data* and how the offloading delay changes as the offloading data increases. We then consider *the bandwidth of the CC*, where the total amount of data and node data are set as constants, and the forwarding delay and link delay are set according to the network topology and the switch performance of the existing network. Finally, we analyzed *the number of nodes*, which shows the performance impact of node size on data offloading.

In each experiment, we randomly varied the set of network topologies and their bandwidths. Here, we only utilized a single CC. A data forwarding link exists between all nodes and the CC, and the link delay between two points reflects the hop count of the nodes on the forwarding link. Transportable links also exist between the nodes.

### 5.3.1. WFO vs. FQ and FCFS: Impact of Data Volume on Delay

We compared the changes in the average delay and data quantity of all the nodes after receiving data under the WFO, WFO(HPC), FQ, and FCFS strategies. Figure 7 shows the experimental results.

The experiment was comprised of four scenarios with different bandwidths and nodes. The path and forward delays are consistent with the experiment described in Section 5.2. The delay in this experiment refers to the time difference between the node receiving data at the beginning and the reception of all data. The delay in Section 5.2 refers to the delay of the data from the CC to a node. The bandwidth of the CC was set to 200 MB/s and 150 MB/s. The number of nodes was set to 100 and 150, respectively. The amount of data from the CC varied from 100 MB to 700 MB.

As depicted in Figure 7, the average delay after offloading data for the three strategies exhibited a linear increase with an increase in data volume. Notably, the WFO strategies (optimal and heuristic) demonstrated lower delays compared to FQ and FCFS strategies. The delay in implementing WFO strategies increased at a slower rate compared to FQ and FCFS strategies which means that as the volume of data increases, the benefits of using WFO become more evident. A lower delay indicates that data can be efficiently distributed across the cloud-edge collaboration network architecture using the WFO strategy. The comparison of FCFS and FQ performances shows sensitivity to the number of nodes while having little impact on bandwidth and data volume. The influence of the number of nodes on WFO performance was relatively small under the same conditions. However, the optimized approach may occasionally result in worse outcomes than the heuristic-based approach. Because both the optimal approach and the heuristic are used to select the

shortest data path in a given queue, in the scenario of this paper, there are multiple queues. The optimal approach must be better than the heuristic for one queue, not necessarily for the whole. Therefore, the optimized approach occasionally reaches worse results than the one using heuristics.

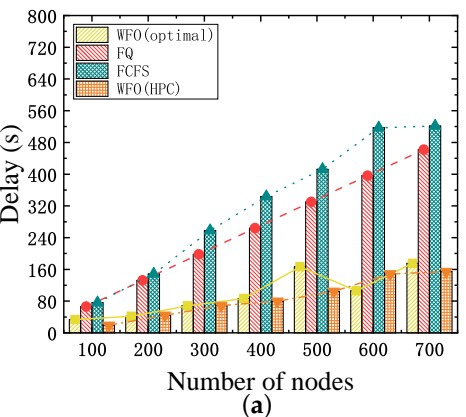 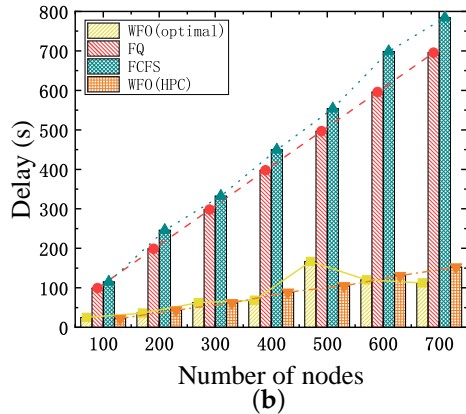

(**a**)  (**b**)

**Figure 7.** WFO vs. other strategies: with the increase of data, the performance advantage of WFO increases gradually. The maximum delay reduction is 73% and 75%, respectively. (**a**) Bandwidth = 150 & Node = 100; (**b**) Bandwidth = 150 & Node = 150.

5.3.2. WFO vs. FQ and FCFS: Impact of the Bandwidth of Cloud on Delay

In this section, the CC bandwidth was used as a variable to measure the delay of various data offloading strategies. The experiment involved 100 and 150 nodes accessing the CC at the same time, while the bandwidth of the CC ranged between 100 MB and 850 MB. The data-tolerance delay was set to 20 ms, and the experiment results are presented in Figure 8.

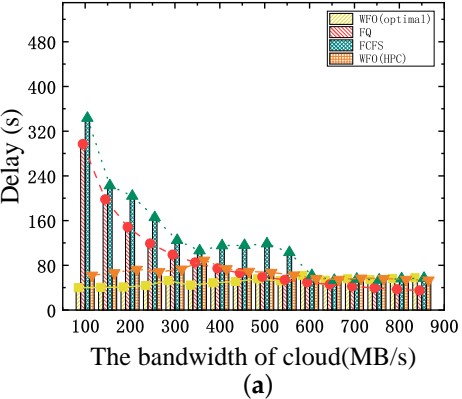 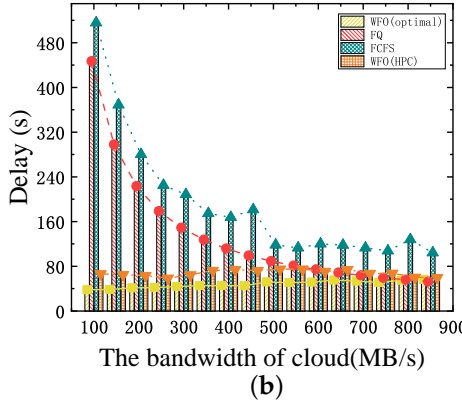

(**a**)  (**b**)

**Figure 8.** WFO can maintain a stable delay under crowded (100–500) and loose (500–900) bandwidth conditions. In the event of congestion, performance increases by 80%. (**a**) Node = 100 & Data = 300; (**b**) Node =150 & Data = 300.

Figure 8 shows the proposed WFO strategy in yellow, the FQ strategy in red, the FCFS strategy in green, and the WFO with HPC in orange. Overall, as the bandwidth of CC increased, the offloading delay of data for all three strategies decreased significantly. The delay for FQ and FCFS decreased significantly, while the delay change for WFO was relatively small due to the increase in the number of nodes accessed simultaneously by FQ and FCFS with the increase in CC bandwidth. The bandwidths of the two nodes gradually met the requirements of all node accesses, resulting in rapid performance improvement. WFO initially performed exceptionally well, with a delay of 78.9 s when the CC bandwidth

was 100 MB/s. FQ and FCFS had delays of 447 s and 515.9 s, respectively. WFO outperformed FQ and FCFS by 82.3% and 84.7%, respectively. Similar results were obtained in other scenarios where an increase in bandwidth led to a slight decrease in WFO. However, the delay was still better than that of FQ and FCFS, which proves that WFO can leverage the characteristics of cloud edge architecture to achieve a stable data offloading capacity. Although WFO has a partial performance loss compared to WFO (optimal), it is still entirely acceptable. Hence, the proposed algorithm HPC holds practical significance.

### 5.3.3. WFO vs. FQ and FCFS: Impact of the Number of Nodes on Delay

This section presents an experiment that examines the impact of the number of ECs on data delay. The experiment was conducted with CC bandwidths of 100 MB/s and 150 MB/s and a total offloaded data amount of 150 MB. The number of nodes ranged from 30 to 140, with a delay tolerance of 20 ms per node. Figure 9 displays the experimental result.

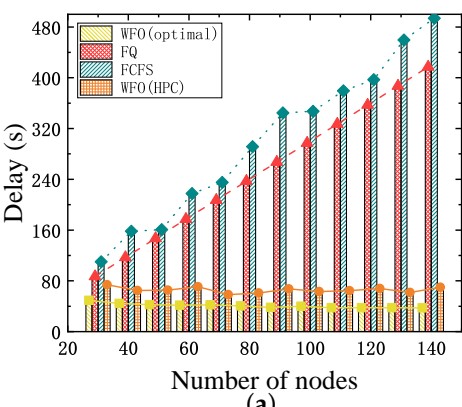 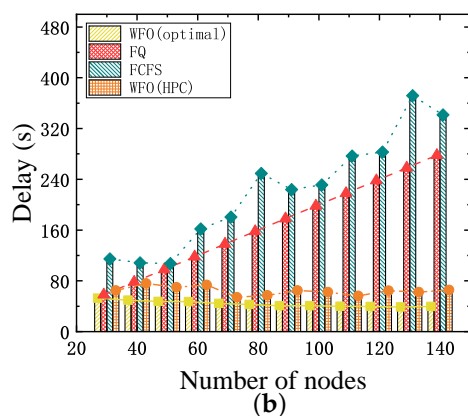

**Figure 9.** The delay is influenced by the number of nodes of the three strategies. The rise in FCFS and FQ is evident. WFO can remain stable. (**a**) Bandwidth = 100 & Date = 150; (**b**) Bandwidth = 150 & Date = 150.

As the number of nodes increased, both FQ and FCFS experienced significant increases in delays, while the delay of WFO remained stable between 60 s and 160 s. When the number of nodes was small, the delays of all three strategies were nearly equal because WFO could not benefit from serving a large number of users. However, it is essential to note that when there are many nodes, WFO can offload data for the same node through multiple nodes, resulting in a more negligible delay than the other two strategies. As the number of nodes increases, the delay of WFO decreases due to its ability to coordinate and significantly reduce the delay of subsequent nodes. Cooperative forwarding allows nodes to lower the entire time delay when calculating the average time delay. Additionally, WFO with HPC performs close to the optimal value, providing evidence of its effectiveness.

### 5.4. The Performance Comparison of WFO and WFO without Priority

When designing the WFO strategy, priority calculation was considered as it can provide significant performance gains. However, it is unnecessary for WFO, and data can be offloaded directly after calculating the groups. Please refer to Figure 10 for more information.

The experimental environment was set up with a CC bandwidth of 300 M/S, a tolerance delay of 20 ms, a data volume of 300 MB, and a range of node numbers 40–140. The results showed that the WFO delay calculated with priority (represented by red) was lower than the delay without priority calculation (represented by green) and the delay with HPC (represented by orange). Therefore, WFO with priority calculation is an effective method to reduce data offloading delays. When there are 120 nodes, the delay for priority calculation with WFO is 38.5 s, while the delay without priority is 77.03 s. This results in a 50.1% reduction in delay. Using WFO with HPC can maintain performance between WFO

with and without priority, making it necessary to calculate priorities. For delay-insensitive data-offloading tasks, WFO with no priority can be used.

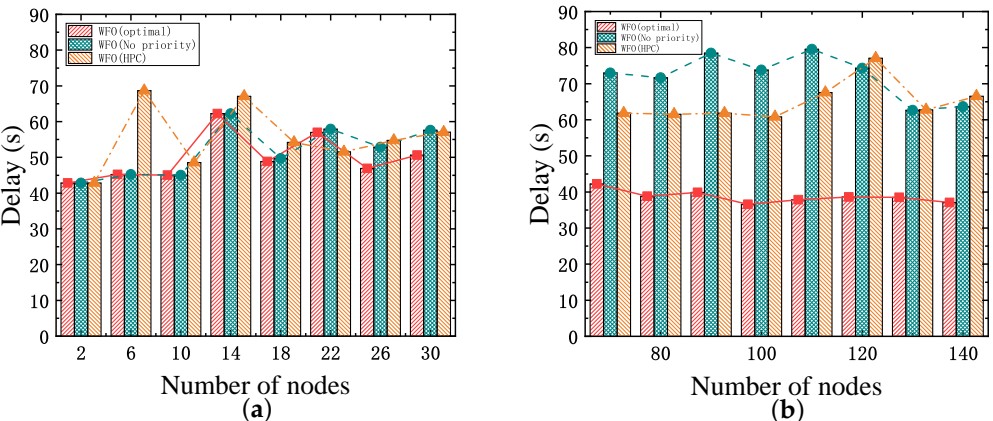

**Figure 10.** The performance gap exists between WFO with priority and WFO without priority. (**a**) Small node topology; (**b**) Large node topology.

*5.5. Delay Dispersion Characteristic*

In practical applications, it is vital to consider data offloading to ensure that the delay difference between nodes requesting data can be managed effectively. Rapid data offloading without delay difference is essential for services requiring cloud-edge collaboration, as individuals in different regions jointly handle these services. The insufficient coordination of data can result in errors in information and eventual service failure. Our study calculated the standard deviation and variance of delay for WFO and FCFS in various CC bandwidth states.

Figure 11 illustrates the delay standard deviation and variance of WFO and FCFS with CC bandwidths ranging from 100–550 M. The blue line represents the variance of WFO, while the red line represents the variance of FCFS. The figure shows that for varying CC bandwidths, the delay variance of WFO is less severe than that of FCFS, which proves that the WFO strategy can effectively reduce the discrete delay between nodes.

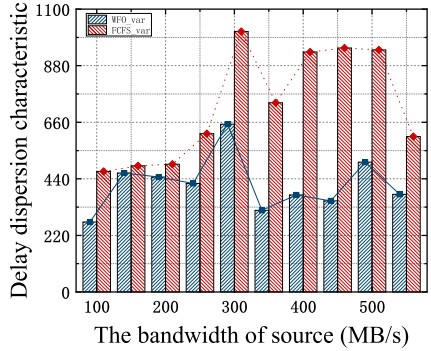

**Figure 11.** Variance and standard deviation of WFO and FCFS conducted on different bandwidths. WFO is better than FCFS in most cases.

## 6. Conclusions

In this paper, we proposed a cloud-edge collaborative data offloading strategy that optimizes the transmission path within a node group to reduce the data offloading delay and balance delay differences between nodes. Our approach leverages the data offloading capabilities of multiple nodes in the cloud and edge computing environments to provide high-quality data services to a more significant number of users despite limited bandwidth. Our experiments demonstrated that WFO could significantly enhance the number of users

accessing the network simultaneously while also improving the service capacity of a data center. Furthermore, our experiments were conducted on networks with an increasing number of nodes, demonstrating the effectiveness of WFO in networks of varying sizes. The performance of WFO was compared to other data offloading strategies, keeping the number of users, data volumes, data center bandwidths, forwarding delays, and link delays constant. The findings revealed that WFO outperformed the other strategies regarding the number of users and latency. Specifically, WFO served up to 80% more users than FQ and FCFS. When the final node has offloaded all data, the resulting delay is up to 80% better than other solutions. WFO can synchronize data between the cloud and the edge. Our future research will concentrate on improving data offloading between terminals and the edge through wireless channels and achieving rapid convergence of data from the edge to the cloud. The former aims to enable efficient data sharing among terminals, while the latter focuses on enabling rapid aggregation capabilities from the edge to the cloud. Additionally, the paper suggests that dynamic networks should be considered when offloading data in a cloud-edge collaboration network.

**Author Contributions:** Conceptualization, S.L., Y.X. and J.X.; Methodology, Z.L. and J.X.; Software, J.Q.; Formal analysis, Y.X.; Writing—original draft, S.L.; Writing—review & editing, S.L.; Supervision, Z.Y. All authors have read and agreed to the published version of the manuscript.

**Funding:** This research received no external funding.

**Institutional Review Board Statement:** Not applicable.

**Informed Consent Statement:** Not applicable.

**Data Availability Statement:** In order to prevent the data from being abused, we open our data based on reasonable requests. Please contact authors with a formal application form to access the data from leeshaonan@outlook.com.

**Conflicts of Interest:** The authors declare no conflict of interest.

## Appendix A

**Remark A1.** *Given a queue $\mathcal{L}'_k$, bandwidth B, and amount of source data S, the transmission delay for Node $\zeta$ in $\mathcal{L}'_2$ to obtain all the data is formulated as:*

$$
t_\zeta = \left( \frac{S}{B_{\zeta=1,C}} + \frac{S}{\left( \sum_{\rho=2}^{\zeta-1} B_{\rho,\zeta} + B_{\zeta,C} - B'_k \right)} \right)
$$
$$
\times \left( 1 - \frac{B'_k}{\sum_{\rho=2}^{\zeta-1} B_{\rho,\zeta} + B_{\zeta,C}} \right)^{\zeta-1} \tag{A1}
$$
$$
- \frac{S}{\left( \sum_{\rho=2}^{\zeta-1} B_{\rho,\zeta} + B_{\zeta,C} - B'_k \right)}
$$

**Proof.** Due to $V_c^\zeta = S - t_{\zeta-1} \times B'_k$, $B'_k = min\{B_{1,\zeta}, B_{2,\zeta}, \ldots, B_{\zeta-1,\zeta}\}$

$$
t_\zeta = t_{\zeta-1} + \frac{S - t_{\zeta-1} \times B'_k}{\sum_{\rho=2}^{\zeta-1} B_{\rho,\zeta} + B_{\zeta,C}} \tag{A2}
$$

Since $\sum_{\rho=2}^{\zeta-1} B_{\rho,\zeta} + B_{\zeta,C}$ is a constant. Let $\theta = \sum_{\rho=2}^{\zeta-1} B_{\rho,\zeta} + B_{\zeta,C}$. By multiplying both sides of this equation by C,

$$
t_\zeta \times \theta = t_{\zeta-1} \times \theta + S - t_{\zeta-1} \times B'_k \tag{A3}
$$

By merging similar terms,

$$
t_\zeta \times \theta = t_{\zeta-1} \times (\theta - B'_k) + S \tag{A4}
$$

Dividing both sides by $\theta$,

$$t_\zeta = t_{\zeta-1} \times \frac{(\theta - B'_k)}{\theta} + \frac{S}{\theta} \tag{A5}$$

Let $\lambda = \frac{(\theta - B'_k)}{\theta}$ and $\epsilon = \frac{S}{\theta}$.

$$t_\zeta = t_{\zeta-1} \times \lambda + \epsilon \tag{A6}$$

Assuming this exists, $\vartheta$ results in Equation (A6) as:

$$t_\zeta + \vartheta = \lambda \times (t_{\zeta-1} + \vartheta) \tag{A7}$$

By setting

$$t_\zeta = \lambda \times t_{\zeta-1} + (\lambda - 1) \times \vartheta \tag{A8}$$

We obtain

$$\vartheta = \frac{\epsilon}{\lambda - 1} \tag{A9}$$

Let $b_\zeta = t_\zeta + \vartheta$

$$b_\zeta = \lambda b_{\zeta-1} \tag{A10}$$

Because $b_1 = \frac{S}{B_{\zeta=1,C}} + \vartheta$

$$b_\zeta = \left( \frac{S}{B_{\zeta=1,C}} + \vartheta \right) \lambda^{\zeta-1} \tag{A11}$$

By substituting $\vartheta, \lambda, b_\zeta = t_\zeta + \vartheta, \epsilon$ into Equation (A11), we obtain:

$$\begin{aligned}
*t_\zeta = \Bigg( &\frac{S}{B_{\zeta=1,C}} + \frac{S}{\left( \sum_{\rho=2}^{\zeta-1} B_{\rho,\zeta} + B_{\zeta,C} - B'_k \right)} \Bigg) \\
&\times \left( 1 - \frac{B'_k}{\sum_{\rho=2}^{\zeta-1} B_{\rho,\zeta} + B_{\zeta,C}} \right)^{\zeta-1} \\
&- \frac{S}{\left( \sum_{\rho=2}^{\zeta-1} B_{\rho,\zeta} + B_{\zeta,C} - B'_k \right)}
\end{aligned} \tag{A12}$$

This concludes the proof. $\square$

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
