# Peer review of "WFO: Cloud-Edge Cooperative Data Offloading Strategy Akin to Water Flow"

_applsci, doi:10.3390/app13105867_

Round 1

Reviewer 1 Report

The paper proposes a cloud-edge collaborative data offloading strategy for cloud computing architecture and cloud-edge collaboration architecture.

The authors mentioned the existing data offloading model could not meet these requirements, and a new data offloading strategy is needed to adapt to cloud-edge collaboration.

The experiments covered various use cases and demonstrated the efficacy and efficiency of the proposed model.

The introduction and abstract need to be fully reviewed, including text organization and sentence structures. The reading became really hard at certain points.

Some comments: from lines 41 to 53 mainly.

“FCFS can meet the bandwidth requirements of all users”.

It represents a limitation, and it is hard to fit users’ needs. So I don’t think this sentence is correct. Usually, saturation leads to long response time, latency, and delay due to the number of users for sure, but also because they move a large amount of data simultaneously.

“Neither of these strategies can be applied to the cloud-edge collaboration network architecture.”

It is interesting to describe why it happens from your point of view.

I’d like to suggest improving the challenge description from lines 55 to 59. It may help in paper contributions.

A simple suggestion

3. SYSTEM MODEL AND PROBLEM -> SYSTEM MODEL AND PROBLEM ‘statement.’

Some considerations

The math model and the proof are excellent contributions to the work. However, it represents a simple proof since the authors consider static approaches only, and networks should not be considered static when edge computing is involved. For instance, delay - Jitter and latency oscillate intensively for data streaming scenarios in edge environments. In addition to that, the throughput from cloud to edge device may represent a representative bottleneck that must be present in the measurements.  

The related work section needs to present an up-to-date state of the art. Currently, it is hard to evaluate the real contribution of this work. I’d like to suggest a complete revision of this section; looking forward to verifying the solutions engaged in recent years in both organizations and academic fields.

Reviewer 2 Report

In this manuscript the authors present an offloading model for cloud-edge architectures. The topic is relevant, but I have some concerns, which should be definitely addressed.

Although the proposed algorithm overperforms the basic ones, the literature is rich with other types of offloading algorithms. Neither Section 2 nor Section 6 justifies why we need another approach. Furthermore Section 6 presenting the related works is too brief, a more comprehensive analysis of the related approaches is required and maybe a table as well to summarise and highlight where the novelty of this approach lies among the similar methods.

It is an interesting phenomenon that occasionally the optimised approach reaches worse results than the one using heuristics (see Figure 7), this should be discussed in more detail. I also have doubts how close this mathematical model is to reality, especially that the motivation brings some really good use cases and examples, but the evaluation parameters seem to be chosen ad-hoc. It is also confusing why the model needs to introduce two types of delays (transmission delay vs. forwarding delay). The source code of the evaluation should be available for reproducibility.

Occasionally the description becomes cursory. 

  • What is the role of virtual machines in Figure 2?

  • “which can be achieved by virtualizing a WFO docker image on the CC” -> this needs more context (connection of the containerization and the mathematical model?)

  • The delay of the WFO strategy was lower than that of the first two strategies, and the increasing trend of the delay was lower than that of the other two strategies” -> in the discussion of the results the readers can easily lose which parts the authors refer to. The manuscript should be checked from that point of view.

  • “As shown in Figure 8, blue is the WFO strategy propose” -> I cannot see any blue..

  • “Because of the high complexity of priority calculation, we propose a heuristic priority calculation algorithm” -> this should be discussed in more detail, e.g. how much does the calculation increase the runtime?  

  • Figure 4 and how exactly those delays were calculated should be discussed in more detail. It is not clear why different forwarding sequences have different data delay in this model.

The algorithm parameters should be listed in a table with a short description as well.

Affiliations seem to be incomplete. 

No precise proofreading of the manuscript has been made. One of the main criteria to accept a manuscript is to be well-written. The entire paper should be checked for typos and grammar mistakes as well. Just a few examples:

  • Section 3 is not mentioned at the end of Section 1

  • (in MB/s)and

  • N6 connect to N5

  • as input the bandwidth Bandwidth between

Reviewer 3 Report

Comments of applsci-2335832 

 The current article proposes a cloud-edge collaborative data offloading strategy, akin to water flow (WFO). When users simultaneously access the same data from the same data source, WFO can provide services for more users within the limited bandwidth of the cloud while ensuring the quality of service. It’s an interesting issue addressed.

1. The authors are suggested to have a thorough review of this work to clearly establish the context of the problem addressed, the starting assumptions of the proposed solution, and an in-depth discussion of the results.

2. The authors should explain in advance the WFO Characteristics shown in Sections 2 and 3.

3. Please clarify the differences between the “serverless” and proposed algorithms of “edge computing and cloud computing”, that is, the advantages and disadvantages, etc.

4. The development of combining cloud-edge collaboration adopted in every aspect of applications is also suggested to include in the survey proposed in the current article.

5. Though, the authors have given the discussion results from data offloading scenario integrated edge computing and cloud computing nodes. However, the mathematical models established for the WFO model need to be provided too.

5. The authors should explain why they didn't adopt other relevant basis techniques.

6. The authors should highlight the contributions of this study in the first section.

7. The authors should give some perspective applications in the future for the proposed model processing with three stages.

8. The future work of this study is suggested to be discussed in the last section.

9. The definition of the adopted variable symbols should be clear, and the suggestions given in proofing the tζ are required many descriptions.

10. There have many sentences with typos, for example, the one shown after Eq. (28), “…substituting …”.

11. The comparison with other works for the exploration of the offloading for WFO is requested included.

12. The contents of the 6th section are suggested move to the 2nd section for the reasons appropriate to the organization of a research article.    

Reviewer 4 Report

The authors present their methodology (WFO) for offloading data from a central server. WFO seeks to cascade edge nodes to improve the data delivery time to the users. The authors present evaluations for different flavours of WFO  in comparison with existing modalities for offloading data.

The paper presents a sound theoretical foundation. However, the math equations are not effectively used. Too many equations that might be better suited for supplemental material. Furthermore, there is a demonstration with no clear statement of what is being demonstrated. Please ensure that demonstrations follow this structure: statement, assumptions, starting point, elaboration, and result (ideally, you arrive at the original statement). 

I have a problem with the choice of terminology: cloud and edge. How the methodology operates could apply to link multiple data centres as well as to link cooperating devices within a network. Cloud services also require load-balancing strategies, and it is irrelevant if the additional node is on-site or in a different data centre. Typically, edge computing refers more to tasks being solved on the user side. But because the authors do not clearly explain the role of the user triggering the request, it is unclear where the edge nodes are. Perhaps illustrations of possible topologies of users accessing a service might clarify the challenges, e.g., multiple users from the same local network, multiple users in various local networks, etc.

Methodologically, WFO seems conceptually similar to location offloading used by streaming services. I think it is essential that the authors clearly state that their approach is meant for live streaming. Otherwise, data localisation would be an indispensable methodology for comparing against.

The assumption that edge devices are server-grade machines does not track with the definition of edge computing. Most likely, the nodes would be the users' machines accessing the service. If we assume that the edge devices are other user systems, e.g., other participants in a video call within the same local network, how does the node load affect WFO? What happens when one of the nodes can no longer cope with the link data throughput? How does bi-directional communication affect the bandwidth? What content knowledge is required to make the cascading model effective in composition scenarios like in the video call scenario, where each node could send its video stream to appear next to everyone else's?

Operational aspects are missing from the paper. Where would the algorithm run? do new clients have to talk to the server in the cloud to get the corresponding node? would a supervisor approach offload the WFO task from the server doing the data offloading and be more responsive to initial requests to connect to the data stream?

Round 2

Reviewer 2 Report

Most of my comments were addressed appropriately, however, based on the authors' response, I have minor concerns about the following:

We know nothing about the simulation environment, its reliability, or the reproducibility of the evaluations. In the response GitHub availability was mentioned, but I did not find any URL in the uploaded version. 

The authors' affiliation still seems to be incorrect, city and country should be included as well.

The latest version of the manuscript (without highlighting the changes) should be also attached. 

Reviewer 3 Report

1. There still exist lots of flaws that are required to revise, for example, the definitions of adopted variables are ambiguous, and the "n" shown in eq. (8) is hard to know without description.

2. What's the mean of "*" used in eq. (15) and (16)?

3. The equation number should be assigned to lines from 416 to 429.

4. Some sections are confusing, for example, the contents shown in the sub-section "5.4. WFO vs. WFO without priority" where is very difficult to grasp. 

5. It is suggested that the final revised version should be provided instead of the version with revised sentences. That is the required final edit with the revised version proved. 

6. The authors are suggested to revise the previously mentioned flaws or the submission should be recommended declined.
